# COPY IS ALL YOU NEED

**Tian Lan**$^{\diamond,\heartsuit,*}$   **Deng Cai**$^{\diamond,*,\dagger}$   **Yan Wang**$^{\diamond,\dagger}$   **Heyan Huang**$^{\heartsuit}$   **Xian-Ling Mao**$^{\heartsuit}$
$^{\diamond}$Tencent AI Lab
$^{\heartsuit}$School of Computer Science and Technology, Beijing Institute of Technology
{lantiangmftby,thisisjcykcd,yanwang.branden}@gmail.com
{hhy63,maoxl}@bit.edu.cn

## ABSTRACT

The dominant text generation models compose the output by sequentially selecting words from a fixed vocabulary. In this paper, we formulate text generation as progressively copying text segments (e.g., words or phrases) from an existing text collection. We compute the contextualized representations of meaningful text segments and index them using efficient vector search toolkits. The task of text generation is then decomposed into a series of copy-and-paste operations: at each time step, we seek suitable text spans from the text collection rather than selecting from a standalone vocabulary. Experiments on the standard language modeling benchmark (WikiText-103) show that our approach achieves better generation quality according to both automatic and human evaluations. Besides, its inference efficiency is comparable to token-level autoregressive models thanks to the reduction of decoding steps. We also show that our approach allows for effective domain adaptation by simply switching to domain-specific text collection without extra training. Finally, we observe that our approach attains additional performance gains by simply scaling up to larger text collections, again without further training.[1]

## 1 INTRODUCTION

Most neural language models (LMs) process text generation tasks by making a series of next-token predictions in an autoregressive manner (Radford et al., 2019; Dai et al., 2019; Khandelwal et al., 2020; Shi et al., 2022). Specifically, LMs generate the next-token distribution over a fixed vocabulary for any given prefix. Then, the next token is selected by a chosen decoding method, such as greedy search and nucleus sampling (Holtzman et al., 2020). This process continues until some stop condition is reached. For example, a special end-of-generation token is emitted, or the generated text reaches the maximum length limit.

Unlike traditional neural language models, we reformulate text generation by copying text segments from existing text collections. The text segments can be of variable lengths, including single words and multi-word phrases. For clarity, we will use the term "phrase" to refer to any contiguous text segments, and a single word can also be seen as a phrase of length 1. We compute a contextualized vector representation for each phrase and pack them into an offline index. At each decoding step, a suitable phrase is retrieved from the offline index and appended to the current prefix. In other words, the next-token predictions in traditional neural language models are replaced by a series of copy-and-paste operations.

Our proposed model, named **CoG** (short for **COPY-GENERATOR**), enjoys the following advantages. First, our method selects *phrases in specific contexts* rather than standalone tokens in a fixed vocabulary. It potentially allows for more accurate candidate representation and selection. Second, our method allows *training-free adaptation* to new knowledge sources because the text collection can be updated in a plug-and-play fashion. It could benefit application scenarios such as domain adaptation and data expansion/filtering. Third, our method allows a sequence of multiple tokens (i.e., multi-word

---

$^{*}$ Contributed Equally.
$^{\dagger}$ Corresponding authors.
[1]Our source codes are publicly available at https://github.com/gmftbyGMFTBY/Copyisallyouneed.

phrase) to be generated in one single step. It could reduce the total number of decoding steps, leading to improved inference efficiency.

We conduct extensive experiments to verify the effectiveness of our proposed CoG. On the standard language modeling benchmark (WikiText-103), our proposed CoG substantially outperforms standard baselines on automatic metrics (26.14 vs. 23.43 MAUVE (Pillutla et al., 2021)) and human evaluation (48% vs. 28% human preference). Moreover, when we directly switch the text collection from the WikiText-103 corpus to a domain-specific corpus, Law-MT (Koehn & Knowles, 2017), our proposed CoG outperforms strong baselines on this domain adaption setting (28.14 vs. 26.85 MAUVE and 52% vs. 36% human preference) without any domain-specific training. Furthermore, when we scale up the text collection of CoG to a larger one, the En-Wiki dataset, we obtain additional gain (26.97 vs. 23.43 MAUVE), again without any further training. Our contributions can be summarized as follows:

- We propose CoG, a method that reformulates text generation tasks as a series of copy-and-paste operations from existing text collections.
- We show that CoG can outperform standard neural language model baselines on existing language modeling benchmarks.
- We demonstrate that CoG allows for training-free adaptations to larger text collections and domain-specific text collections.

## 2 BACKGROUND: NEURAL TEXT GENERATION

Neural text generation can be divided into two categories: (1) unconditional text generation; (2) conditional text generation. Unconditional text generation (or language modeling) aims to generate a coherent text continuation given a prefix. In this case, language models perform generation using a density estimation over sequences $p_\theta(x)$. Conditional text generation aims to generate text with some condition $c$ and instead estimates the probability of $p_\theta(x|c)$. Its typical applications include machine translation (Sutskever et al., 2014; Bahdanau et al., 2015), summarization (See et al., 2017). Throughout this paper, our discussion will be focused on unconditional text generation, however, our approach can be readily adapted to conditional text generation as well.

The canonical approach to language modeling factors the generation in an autoregressive left-to-right manner $p_\theta(x_{0:n}) = \prod_{i=1}^{n} p(x_i|x_{<i})$. In this case, text generation is reduced to the task of repeatedly predicting the next token conditioned on the partial sequence (i.e., prefix) generated so far $p(x_i|x_{<i})$. The model often consists of two parts: (1) a prefix encoder and (2) a set of token embeddings. The prefix encoder is often parameterized by the Transformer architecture (Vaswani et al., 2017), which transforms any prefix into a fixed-sized vector representation $h_i \in \mathbb{R}^d = \text{PrefixEncoder}(x_{<i})$. Then, the probability of the next token being $w$ is calculated as

$$p_\theta(x_i = w|x_{<i}) = \frac{\exp(v_w \cdot h_i)}{\sum_{w \in V} \exp(v_w \cdot h_i)},$$

where $v_w$ is the context-independent token embedding representing the token $w$, and $V$ is the predefined vocabulary consisting of all possible tokens. Based on the chosen decoding method, such as greedy search and nucleus sampling (Holtzman et al., 2020), the next token is selected according to the probability distribution over the fixed vocabulary $V$. This process is repeated in an autoregressive manner, until some stop condition is reached, e.g., the maximum length of generation is reached.

## 3 COPY-GENERATOR

Unlike traditional language models that compute the next token distribution over a fixed vocabulary that is usually composed of words or sub-words (Sennrich et al., 2016; Kudo & Richardson, 2018), our proposed CoG has a dynamic "vocabulary" that is dependent on the available source text collections. Each item in the "vocabulary" corresponds to a text segment (termed as *phrase* in this paper) in the source text collection. Importantly, all phrases are context-sensitive. That is, the same phrases in different contexts are considered to be different. The overall framework is depicted in Figure 1.

Formally, our approach assumes a set of source documents $\{D^1, \ldots, D^n\}$ is available. For each document $D^i$, a phrase $k = D^i_{s:e}$ of length $e - s + 1$ can be extracted, where $s$ and $e$ mark the start

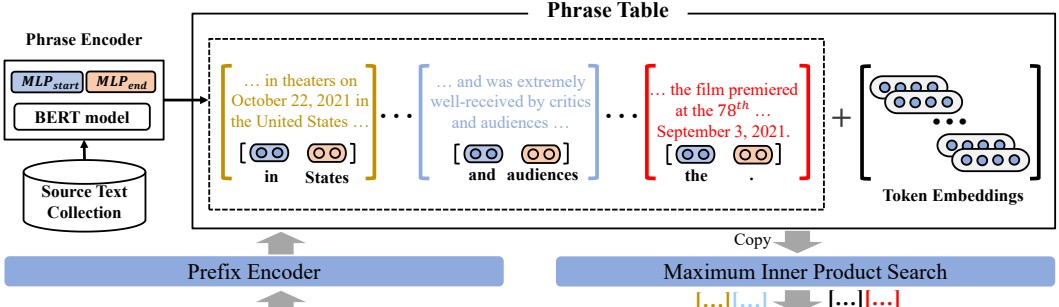

The Dune film was released [in theaters on October 22, 2021 in the United States] [and was extremely well-received by critics and audiences] [Before] [that] [,] [the film premiered at the $78^{th}$ International Film Festival on September 3, 2021.]

Figure 1: The overview of our proposed CoG. Given the prefix *The Dune film was released*, CoG retrieve 3 phrases (in different colors) from the documents and generates 3 tokens (*Before*, *that*, and the comma ,) from the fixed vocabulary to form the whole generation.

and end positions of the phrase in the document, respectively. We denote all the phrases in the source text collection as $\mathcal{P}$. For a given prefix $x_{<i}$, we aim to select the best phrases that can form a coherent text continuation following the prefix. To this end, we compute a contextualized representation for each phrase $p_k \in \mathbb{R}^d = \text{PhraseEncoder}(s, e, D^i)$ using a phrase encoder. Thus, a phrase table $\{(k, p_k)|k \in \mathcal{P}\}$ can be constructed. Similar to traditional language models, at test time, CoG also employs a prefix encoder to map the prefix $x_{<i}$ into a vector representation $q_i$. The fitness of a phrase $k$ to the prefix $x_{<i}$ is then measured by the dot product of their vector representations $p_k$ and $q_i$:

$$p(k|x_{<i}) \propto \exp(p_k \cdot q_i). \tag{1}$$

At each time step, a suitable phrase is selected and appended to the current prefix accordingly.

Note that the size of the phrase table can be up to billions. To search over this large candidate pool, we pre-compute the phrase representations and use a coarse-to-fine search pipeline based on maximum inner product search (MIPS) (Johnson et al., 2019). The details are deferred to Section 4.2. Moreover, to support the scenarios where no suitable phrases are available, we also add the context-independent token embeddings $\{(w, v_w)|w \in V\}$ in standard LMs to the phrase table.

**Ethical Consideration**  The text generated by CoG contains text segments copied from other documents, which may cause copyright disputes in real-world applications. Therefore, there are a few things to be considered: (1) The copyright of the source text documents needs to be carefully checked. One should not use documents with strict copyright protection and/or private information; (2) It is recommended to quote the original source explicitly, especially when the retrieved phrases are long.

### 3.1 MODEL ARCHITECTURE

As illustrated in Figure 1, our proposed model consists of three major components: (1) a *prefix encoder* that maps prefixes to fixed-sized representations; (2) a context-dependent *phrase encoder* that computes the vector representations of the phrases in the source text collection; (3) a set of context-independent *token embeddings* similar to the one used in standard neural language models.

**Prefix Encoder**  The prefix encoder is responsible for encoding the prefix $x_{<i}$ into a vector representation for the next-phrase prediction. We treat the prefix as a sequence of tokens (previously predicted phrases are split into tokens as well) and encode them using the standard Transformer architecture with causal attention (Vaswani et al., 2017; Radford et al., 2019). Causal attention only allows each position in the input sequence to attend to its preceding positions. Therefore, the prefix representation can be computed incrementally as the generation progresses, leading to faster inference. Concretely, the prefix encoder transforms a prefix $x_{<i}$ of length $i$ into a matrix $\mathcal{H}_i \in \mathbb{R}^{i \times dL}$, where $d$ is the hidden dimension and $L$ is the number of Transformer layers. The computation can be written as:

$$\mathcal{H}_{i+1} = \text{PrefixEncoder}(x_i, \mathcal{H}_i).$$

We use the hidden state of the last token as the prefix representation $q_i$.

**Phrase Encoder**    Given a set of source documents $\{D^1, ..., D^n\}$, the phrase encoder computes the vector representations of all the phrases in the documents. Inspired by previous work (Lee et al., 2016; Seo et al., 2018; Lee et al., 2021), we construct context-dependent phrase representations as follows. For a document $D = D_1, \ldots, D_m$ of length $m$, we first apply a deep bidirectional Transformer (Devlin et al., 2019) to obtain contextualized token representations $\mathcal{D} \in \mathbb{R}^{m \times d_t}$, where $d_t$ is the dimension of token representations. Then, we apply two MLPs models, $\text{MLP}_{\text{start}}$ and $\text{MLP}_{\text{end}}$, to convert $\mathcal{D}$ into start and end token representations $\mathcal{D}_{\text{start}}, \mathcal{D}_{\text{end}} \in \mathbb{R}^{m \times \frac{d}{2}}$, respectively:

$$\mathcal{D}_{\text{start}} = \text{MLP}_{\text{start}}(\mathcal{D}), \mathcal{D}_{\text{end}} = \text{MLP}_{\text{end}}(\mathcal{D}).$$

For each phrase $D_{s:e}$ that starts at $s$ and ends at $e$ in the document, we use the concatenation of the corresponding start and end vectors as the phrase representation.

$$\text{PhraseEncoder}(s, e, D) = [\mathcal{D}_{\text{start}}[s]; \mathcal{D}_{\text{end}}[e]] \in \mathbb{R}^d \tag{2}$$

The advantages of the above representation method are that (1) we only need to encode the document once to obtain all phrase representations; and (2) we only need to store all the token representations instead of all phrase representations.

**Context-Independent Token Embeddings**    Although CoG can copy phrases from other documents, we would like to retain the generalization capability to compose output with standalone tokens. This can be especially useful when there is no suitable phrase in the source text collection. Therefore, we also add the traditional context-independent token embeddings $\mathcal{V} \in \mathbb{R}^{|V| \times d}$ to our phrase table. These tokens can be seen as phrases of length 1 without any context information.

## 3.2 MODEL TRAINING

CoG decomposes the task of text generation into a series of copy-and-paste operations: at each time step, it selects the next phrase either from the source text collection or the fixed token vocabulary. In other words, phrases are used as the basic building blocks for text generation. To train CoG, each document in the training set is chunked into a sequence of phrases in a similar spirit. Specifically, we propose a greedy segmentation algorithm based on forward maximum matching. Taking a document $D = D_1, \ldots, D_m$ of $m$ tokens as an example, our algorithm segments the document from left to right. The first $i$ tokens will be cut out as a phrase if it can be found as a sub-sequence in other documents and $i$ is the maximum valid value. The above process is repeated until all tokens are cut out. Note that some resultant phrases can be single tokens in the fixed token vocabulary when no proper matching can be found. Detailed explanations of the phrase segmentation algorithm can be found in Appendix D.

Suppose that a document $D$ has been split into $n$ phrases $D = p_1, \ldots, p_n$. If the $k$-th phrase $p_k$ is copied from another document, let $D^k$ be the source document and let $s_k, e_k$ be the start and end positions of $p_k$ in $D^k$, the phrase encoder is used to extract its context-dependent phrase representations $\text{PhraseEncoder}(s_k, e_k, D^k)$ (Eq. 2). On the other hand, we directly retrieve the context-independent token embedding of $p_k$ if it is copied from the fixed token vocabulary. As illustrated by Eq. 1, CoG relies on a shared vector space of prefix and phrase representations, where the representations of semantically coherent prefixes and phrases should be closer to each other while others should be pushed apart. We define the training loss for next-phrase predictions by using the InfoNCE loss with in-batch negatives (Karpukhin et al., 2020):

$$\mathcal{L}_p = -\frac{1}{n} \sum_{k=1}^{n} \log \frac{\exp(q_k \cdot p_k)}{\sum_{p \in \mathcal{P}_k} \exp(q_k \cdot p_p) + \sum_{w \in V} \exp(q_k \cdot v_w)}$$

where $\mathcal{P}_k$ consists of all the phrases in the source document $D^k$, $V$ is the set of all tokens in the token vocabulary, and $q_k$ denotes the representation of the prefix preceding the phrase $p_k$ in $D$.

Additionally, to retain the capability of token-level generation, we also train CoG with the standard token-level autoregressive loss.

$$\mathcal{L}_t = -\frac{1}{m} \sum_{i=1}^{m} \log \frac{\exp(q_i, v_{D_i})}{\sum_{w \in V} \exp(q_i, v_w)}$$

where $q_i$ denotes the prefix representation preceding the token $D_i$ in $D$. Finally, the training loss is the sum of these two losses:

$$\mathcal{L} = \mathcal{L}_p + \mathcal{L}_t$$

## 4 EXPERIMENTAL SETUP

### 4.1 BASELINES

We compare CoG with the following three baselines:

- **Transformer** (Vaswani et al., 2017) has been the *de facto* model for neural language models. Concretely, we fine-tune the pre-trained GPT2 model (Radford et al., 2019) in our experiments.
- $k$**NN-LM** (Khandelwal et al., 2020) is a retrieval-augmented generation model, which extends a pre-trained neural language model by linearly interpolating its next token distribution with a $k$-nearest neighbors ($k$NN) model.
- **RETRO** (Borgeaud et al., 2022)[2] is another retrieval-augmented generation model which combines a frozen BERT retriever, a differentiable encoder and a chunked cross-attention mechanism to predict next tokens. Since there is no pre-trained RETRO model that could be accessed, we train it from scratch on the WikiText-103 dataset.

### 4.2 IMPLEMENTATION DETAILS

All the baselines and our source codes are based on the popular Huggingface transformers package (Wolf et al., 2020). For a fair comparison, the prefix encoders in Transformer, $k$NN-LM, and CoG use the same model architecture as the pre-trained GPT2 model (12 layers, 12 heads, and 768 hidden dimensions) (Radford et al., 2019). For the phrase encoder in CoG, we fine-tune the pre-trained BERT-base-cased model (Devlin et al., 2019) (12 layers, 12 heads, and 768 hidden dimensions). We train baselines and CoG for 400,000 steps on 8 Tesla-V100 GPUs. For all the baselines, the learning rate, dropout rate, and gradient clipping are set as 5e-5, 0.1, and 1.0, respectively. Due to memory limitation, the batch size is set to contain 256 phrases. For the BERT model in the phrase encoder, the maximum sequence length is set as 256. For the GPT2 model in the prefix encoder, the maximum sequence length is set as 512. Our proposed CoG contains overall 248M parameters from BERT and GPT2 models, and other baselines contain over 124M parameters. As suggested by Borgeaud et al. (2022), the hyper-parameters $\lambda$ and $\alpha$ of $k$NN-LM are set as 0.118 and 0.00785, respectively.

To improve the inference efficiency of CoG, we encode all the documents in the source text collections offline. Note that retrieving from such a super large phrase collection faces severe challenges on the engineering side. This paper uses a coarse-to-fine pipeline to address this challenge. Specifically, we first use a document retriever to retrieve top-$k$ related documents for each given prefix. Then, their corresponding phrase representations are collected for selection. In this paper, a popular semantic matching model, DPR (Karpukhin et al., 2020) and a vector search toolkit, FAISS (Johnson et al., 2019) are used as the document retriever, which can recall documents that have similar topics with the prefix. The value $k$ is empirically set to 1024.

CoG can be used with both greedy search and nucleus sampling. For greedy search, CoG selects the phrase that has the highest fitness score at each time step. As for nucleus sampling, we first obtain the next-phrase distribution by using the `softmax` function over the fitness scores of all candidate phrases. Then, the next phrase is sampled over this distribution.

More details of the implementation can be found in Appendix A and B.

### 4.3 AUTOMATIC EVALUATION METRICS

For each document in the test set, we use the first 32 tokens as the prefix. The baselines and our proposed CoG generate text continuations of length 128 based on the same prefix. Following

---

[2]https://github.com/lucidrains/RETRO-pytorch.

conventions (Welleck et al., 2020; Su et al., 2022), we use **greedy search** and **nucleus sampling** (Holtzman et al., 2020) ($p = 0.95$) throughout our experiments. Following previous work (Welleck et al., 2020; Su et al., 2022) and report the results on the following evaluation metrics:

- **MAUVE** (Pillutla et al., 2021), an efficient, interpretable, practical automatic evaluation, is highly coherent with human judgments and widely used to evaluate modern text generation models (Su et al., 2022; Krishna et al., 2022). In this paper, MAUVE leverages the GPT2-large model to generate the scores, and the scaling factor is set as 2.0.

- **Rep-$n$** (Welleck et al., 2020) measures the sequence-level repetition as the portion of duplicate n-grams in the generated text (Welleck et al., 2020). For a generation text $x$, Rep-$n$ can be formulated as: $100 \times (1.0 - \frac{|\text{unique n}-\text{gram(x)}|}{|\text{total n}-\text{gram(x)}|})$. Higher Rep-$n$ denotes the severe degeneration problem in generations.

- **Diversity** (Welleck et al., 2020) measures the diversity of the generations, which is formulated as $\Pi_{n=2}^{4}(1 - \frac{\text{Rep}-n}{100})$. Generations that have higher Diversity scores usually are more informative.

Note that previous work (Khandelwal et al., 2020; Dai et al., 2019) often uses perplexity as the primary evaluation metric to measure the performance of language modeling. However, since our proposed CoG does not calculate next-token distributions over a fixed vocabulary, the comparison of perplexities is not reliable and thus omitted. However, we can test the perplexity of generated text using an external language model, and the results are shown in Appendix C.

## 5 EXPERIMENTAL RESULTS

In this paper, we evaluate baselines and our proposed CoG in three different settings: (1) standard language modeling; (2) domain adaption; (3) enlarged phrase index.

### 5.1 LANGUAGE MODELLING ON WIKITEXT-103

In this setting, models are trained on the training set of the WikiText-103 dataset and evaluated on its test set. The WikiText-103 dataset (Merity et al., 2017) contains an extensive collection of Wikipedia articles with over 100 million words, which is widely used to evaluate the performance of universal language modeling (Khandelwal et al., 2020; Dai et al., 2019; Su et al., 2022).

| Model | Decoding | MAUVE↑ | Rep-2↓ | Rep-3↓ | Rep-4↓ | Diversity↑ | Latency (s)↓ |
|---|---|---|---|---|---|---|---|
| **Transformer** | greedy | 19.87 | 43.56 | 38.55 | 35.5 | 22.37 | 1.32 |
| | nucleus | 23.43 | 5.10 | 1.33 | 0.50 | 93.22 | 1.48 |
| $k$**NN-LM** | greedy | 19.92 | 43.79 | 38.76 | 35.69 | 22.13 | 10.36 |
| | nucleus | 22.50 | **3.33** | **0.69** | **0.21** | **95.8** | 10.42 |
| **RETRO** | greedy | 21.19 | 44.65 | 39.63 | 36.6 | 21.19 | 4.39 |
| | nucleus | 22.86 | 6.21 | 1.93 | 0.86 | 91.19 | 4.51 |
| **CoG** | greedy | 26.01 | 28.14 | 23.80 | 21.40 | 43.03 | **1.29** |
| | nucleus | **26.14** | 7.31 | 2.66 | 1.28 | 89.07 | 1.54 |

Table 1: The automatic evaluation on the test set of WikiText-103. As for each model with nucleus sampling, we run 10 times and recorded the average MAUVE and Diversity scores.

**Results**   Table 1 shows the performance comparison between the baselines and our proposed CoG on the test set of the WikiText-103 corpus. It can be found that our proposed CoG substantially outperforms the Transformer and $k$NN-LM baselines on most metrics. Specifically, CoG improves MAUVE score over the best baseline (Transformer with nucleus sampling) from 23.43 to 26.14 – an improvement of 2.71%. Interestingly, although it is well known that greedy search could raise severe degeneration problems (Welleck et al., 2020), CoG with greedy search still outperforms the standard Transformer baseline with nucleus sampling, with 2.58% improvements on MAUVE. This

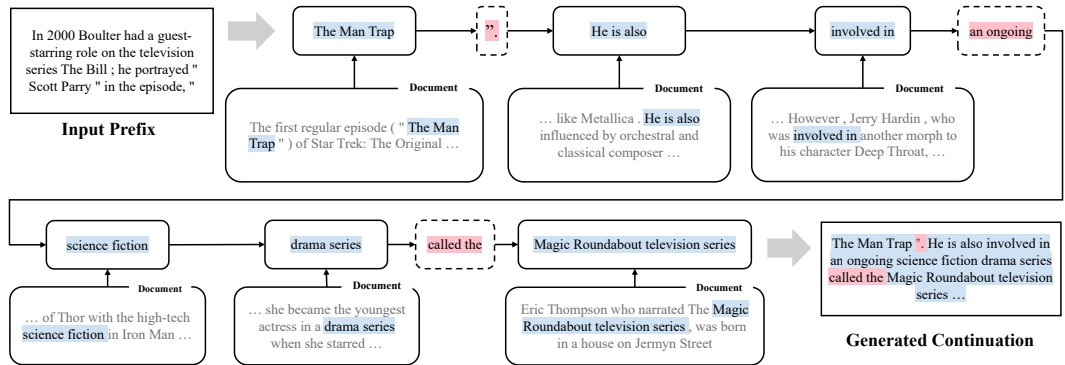

Figure 2: An example generated by CoG on the test set of WikiText-103. The dotted squares denote that the content (highlighted in red )is copied from the token vocabulary, and the solid squares denote that the content (highlighted in blue ) is copied from other documents.

observation demonstrates that CoG is more robust and less prone to the degeneration problem, which can be considered as an additional bonus.

**Inference Speed** Furthermore, we also compare the average time cost of different methods for completing the generation on the test set. Since the phrase representations in CoG are pre-computed offline, its encoding time cost is not included. The results are reported in Table 1. As seen, CoG still achieves comparable inference efficiency with the standard Transformer baseline. The reason is that the copied phrases usually contain multiple tokens (the statistics of phrase length are shown in Table 2). As a result, CoG uses fewer decoding steps when generating the text of the same length. Unlike CoG that uses a coarse-to-fine search pipeline, $k$NN-LM conducts large-scale vector search at every decoding step. Its inference latency is much higher than Transformer, and CoG, which is aligned with previous work(Alon et al., 2022).

| Method | Uni-gram | 2-gram | 3-gram | 4-gram | 5-gram | 6-gram |
|---|---|---|---|---|---|---|
| Greedy | 0.583 | 0.195 | 0.121 | 0.056 | 0.029 | 0.017 |
| Nucleus | 0.434 | 0.219 | 0.181 | 0.09 | 0.048 | 0.028 |

Table 2: The statistics on the length of the copied phrases (on the test set of WikiText-103).

**Human Evaluation** To ensure the reliability of our evaluations, we also run human evaluation with three native-speaker graders from a third-party grading platform. Specifically, we randomly select 100 test prompts. For each test prompt, the annotators are given two continuations, in random order, which are generated by CoG and Transformer respectively. The annotators are asked to decide which one is better by considering the following aspects:

| Comparison | Better | No Prefer. | Worse |
|---|---|---|---|
| CoG vs. Transformer | **48**% | 24% | 28% |

Table 3: Human evaluation on the WikiText-103 corpus.

- **Fluency**: Whether the generated text is fluent and easy to understand.
- **Informativeness**: Whether the generated text is diverse and contains interesting content.

When annotators make different decisions on the same sample, we ask them to have a discussion and make the final decision. As shown in Table 3, our proposed CoG model significantly outperforms strong Transformer baseline, indicating its better generation quality.

**Case Study** For a better understanding of the performance of CoG, we present an example of the text continuations generated by our proposed CoG in Figure 2. It can be found that CoG can retrieve phrases that are semantically coherent and fluent for given prefixes. For example, at the second decoding step, CoG generate the punctuations ["", .] from the pre-defined vocabulary to close the film name *"The Man Trap"* and the sentence. Besides, at the ninth decoding step, CoG directly copied the named entity *Magic Roundabout television series* from the related document. More examples can be found in Appendix E.

## 5.2 DOMAIN ADAPTION ON LAW-MT

In the domain adaption setting, the models trained on the WikiText-103 dataset are tested on a specific domain. Following previous work (He et al., 2021; Alon et al., 2022), we use the English part of Law-MT (Koehn & Knowles, 2017), which is an English-German translation dataset for law documents. The memory of $k$NN-LM, RETRO and CoG are constructed from the training set of Law-MT. We also present the performance of Transformer baselines with or without further fine-tuning on the training set of Law-MT.

**Results**  As shown in Table 4, it can be observed that CoG even outperforms the Transformer model further fine-tuned on the Law-MT corpus (Transformer w/ FT). Specifically, CoG outperforms Transformer w/ FT by 2.93% MAUVE score. The results indicate that CoG allows a single model to be specialized in different domains, by simply switching the source text collection. Although $k$NN-LM brings in higher Diversity scores, CoG surpasses it by 3.39% MAUVE score, which shows CoG has higher generation quality in general.

| Model | Decoding | MAUVE ↑ | Diversity ↑ |
|---|---|---|---|
| Transformer w/o FT | greedy | 20.32 | 70.66 |
| | nucleus | 25.21 | 93.88 |
| Transformer w/ FT | greedy | 23.00 | 80.52 |
| | nucleus | 26.85 | 90.14 |
| $k$NN-LM | greedy | 23.31 | 19.85 |
| | nucleus | 24.75 | **94.60** |
| RETRO | greedy | 18.70 | 71.14 |
| | nucleus | 20.35 | 94.81 |
| CoG | greedy | 21.31 | 84.32 |
| | nucleus | **28.14** | 92.56 |

Table 4: The automatic evaluation on Law-MT.

**Human Evaluation**  We also conduct the human evaluation on the Law-MT corpus, which has a similar setup to that in (§5.1). Table 5 shows that most of CoG's generations are better than a strong Transformer baseline. This observation demonstrates that CoG can even

| Comparison | Better | No Prefer. | Worse |
|---|---|---|---|
| CoG vs. Transformer w/ FT | **52%** | 12% | 36% |

Table 5: Human evaluation on Law-MT.

outperform the fine-tuned Transformer baseline without any domain-specific training.

## 5.3 ENLARGED PHRASE INDEX WITH EN-WIKI

In the enlarged phrase index setting, we make use of a large text collection, the En-Wiki corpus, and test baselines on the test set of WikiText-103. The En-Wiki corpus contains a large-scale collection of Wikipedia articles with over 3 billion words, whose size is much larger than the WikiText-103 dataset. The memory of $k$NN-LM, RETRO, and CoG are built from the training set of En-Wiki[3]. Similar to the domain adaption setting, we also present the results of Transformer baselines with or without further fine-tuning on the En-Wiki corpus.

**Results**  The experimental results are shown in Table 6. CoG with En-Wiki memory surpasses other strong baselines and CoG with WikiText-103 memory. This is especially remarkable because CoG does not require any additional training, suggesting we can train CoG with a smaller corpus but leverage additional information in a larger corpus in a plug-and-play fashion. Similar to the domain adaption setting, we also notice that, although $k$NN-LM baseline improves Diversity scores, it obtains a much lower MAUVE score than CoG (23.39 vs. 26.97). Note that the Transformer w/ FT is slightly worse than that without fine-tuning on the En-Wiki dataset. This phenomenon is mainly because there are deviations between En-Wiki and WikiText-103 datasets.

**Effects of Index Size**  To further investigate how the size of the phrase index affects the generation quality, we randomly sample several subsets of the En-Wiki dataset with proportions from $0.1\%$ to $100\%$. As shown in Figure 3, when the proportion is less than $1\%$, **CoG** exhibits a similar quality, which is unsurprising since few enlarged documents are added to the phrase index. In contrast, once the proportion is larger than $1\%$, the larger the phrase index becomes, the better generation quality the model achieves.

---

[3]Due to the hardware limitation, RETRO uses the subset of the En-Wiki corpus (over 6 million chunks).

| Model | Decoding | MAUVE ↑ | Diversity ↑ |
|---|---|---|---|
| **Transformer w/o FT** | greedy | 19.87 | 22.37 |
| | nucleus | 23.43 | 93.22 |
| **Transformer w/ FT** | greedy | 20.21 | 19.62 |
| | nucleus | 21.31 | 92.92 |
| $k$**NN-LM** | greedy | 23.21 | 20.33 |
| | nucleus | 23.39 | **96.37** |
| **RETRO** | greedy | 19.75 | 21.15 |
| | nucleus | 22.87 | 91.09 |
| **CoG** | greedy | 24.68 | 40.45 |
| | nucleus | **26.97** | 90.00 |

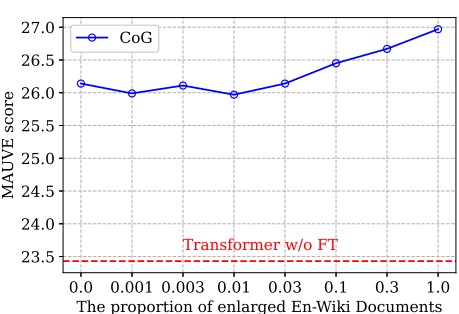

Table 6: The automatic evaluation on the test set of WikiText-103, the memory is built on the train set of En-Wiki. Transformer w/ FT and Transformer w/o FT denote the Transformer baseline with and without further fine-tuning on the train set of En-Wiki, respectively.

Figure 3: Generation quality of **CoG** with different sizes of the phrase index. For each proportion (point in the X-axis), we sample 10 times and record the averaged MAUVE score. A proportion of 0.0 indicates that only documents from WikiText-103 are used.

## 6    RELATED WORK

**Dense Retrieval**    The dense retrieval technique (Karpukhin et al., 2020) has been widely used in many downstream NLP tasks, such as open-domain question answering (Karpukhin et al., 2020; Lee et al., 2021), open-domain dialogue systems (Lan et al., 2021) and machine translation (Cai et al., 2021). Different from the traditional sparse retrieval system, such as BM25 and TF-IDF (Robertson & Zaragoza, 2009), dense retrieval learns a shared vector space for queries and documents, where relevant pairs of query and document have smaller distances (i.e., higher similarities) than the irrelevant pairs.

The most closely related work to our study is DensePhrase (Lee et al., 2021). DensePhrase reformulates the question-answering task as a phrase retrieval problem, where phrases are directly retrieved and returned as answers to factual questions. Differently, our work aims to generate coherent text continuations through multiple rounds of phrase retrieval. Since the connection between two adjacent phrases should be coherent and fluent in the text generation task, it is much more difficult.

**Retrieval-Augmented Text Generation (RAG)**    Retrieval-augmented text generation has gained increasing interest recently. Most prior work improves the generation quality (e.g., informativeness) of language models by grounding the generation on a set of retrieved materials (e.g., relevant documents) (Li et al., 2022; Guu et al., 2020; Hashimoto et al., 2018; Weston et al., 2018; Cai et al., 2019a;b; Khandelwal et al., 2020; Wu et al., 2019; Guu et al., 2020; Lewis et al., 2020; Borgeaud et al., 2022; Yang et al., 2023). Our work is on this line of research but takes a radical step forward. Unlike prior work that builds the combinations of retrieval and generation, retrieval is generation in CoG.

One contemporary work to our work is Min et al. (2022), which shares the idea of replacing the fixed vocabulary with a nonparametric phrase table. However, Min et al. (2022) focuses on masked language modeling while our focus is on causal language modeling and text generation.

## 7    CONCLUSION

In this paper, we reformulated text generation as progressively copying phrases from the massive text collection. Following this formalization, we proposed a novel neural text generation model, named CoG, which generates text by retrieving semantically coherent and fluent phrases from other documents. Experimental results proved the advantages of CoG over the strong baselines on three experimental settings: standard language modeling (WikiText-103), domain adaptation (Law-MT), and enlarged phrase index (En-Wiki).

## JUSTIFICATION OF CHANGES

Note that the experimental results in the current version have some changes from the previous version that has been reviewed. We made a number of revisions to the experiments according to the valuable suggestions from the reviewers.

## ACKNOWLEDGEMENT

The authors thank the anonymous reviewers for their valuable suggestions and comments on our paper, which significantly improves the quality of our paper.

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

## A  DATASET STATISTICS

The experiments in this paper include three benchmarks: (1) WikiText-103; (2) English part of Law-MT; (3) En-Wiki. The statistics of these benchmarks are shown in Table 7. En-Wiki corpus is used for the enlarged phrase index settings in this paper, containing over 4,848,348 long English Wikipedia documents.

| Benchmarks | Train | Dev | Test |
|---|---|---|---|
| **WikiText-103** | 1,801,350 | 3,760 | 4,358 |
| **Law-MT** | 389,292 | 2,000 | 2,000 |

Table 7: The number of sentences in the WikiText-103 and Law-MT datasets.

## B    MORE IMPLEMENTATION DETAILS

During training, the dynamic vocabulary of CoG contains two parts: (1) word-level vocabulary size (50257 in GPT2 vocabulary); (2) the phrases in a batch of training documents. During inference, the dynamic vocabulary consists of the word-level vocabulary and the phrases extracted from the Top-$k$ retrieved documents ($k$=1024 in this paper). The size of the pre-defined word-level vocabulary contains 50257 subwords. Since there are only a few documents encoded to extract the phrase representations, the average number of the phrase representations is 950,942.4 in the WikiText-103 test set when $K = 1024$.

## C    PERPLEXITY OF GENERATED TEXT

We calculate the perplexity of the generated texts under a large pre-trained language model (GPT2-Large). As shown in Table 8, it can be found texts generated by greedy search can achieve very low perplexity scores (even much lower than the ground-truth)[4]. This is expected as greedy search targets at likelihood maximization. Sampling-based decoding methods give much higher perplexity scores. Moreover, it is worth noting that CoG achieves the closest perplexity score to ground-truth.

| Models | Perplexity | |
|---|---|---|
| | greedy | nucleus |
| **Transformer** | 3.26 | 37.11 |
| $k$**NN-LM** | 3.48 | 78.01 |
| **RETRO** | 3.27 | 36.40 |
| **CoG** | 10.41 | 27.24 |
| **Ground-Truth** | 18.64 | |

Table 8: The perplexity on the test set of WikiText-103.

## D    THE PHRASE SEGMENTATION ALGORITHM

CoG takes phrases as the minimum units that can be put together to form a coherent document. To train CoG, we design a phrase segmentation algorithm to split each document in the training set into a sequence of phrases. This algorithm makes use of a forward maximum matching strategy to identify phrases. Maximum matching is one of the most popular structural segmentation algorithms. This method favors long phrases and is a greedy algorithm by design. Specifically, we treat each document as a sequence of tokens and scan each document from left to right. At each step, we search for the longest prefix of the unsegmented part that is also a sub-sequence of other documents other than the current document. If the length of that prefix is bigger than 2, we take that prefix as the next phrase. Otherwise, we take the first token as the next phrase and it is labeled as coming from the fixed token vocabulary. In both cases, we process the rest part of the current document recurrently. The algorithm can be very time-consuming because exhaustive searches over millions of documents are compute-intensive. Therefore, we propose an efficient approximation as follows. First, we retrieve the top-$k$ most similar documents for each document using the popular DPR model (Karpukhin et al., 2020)[5], and vector search toolkits, FAISS (Johnson et al., 2019). Then, the phrase search only runs on the corresponding top-$k$ documents. The relevant documents usually have similar topics to the current document. The value of $k$ is set as 1024 in our experiments. The details of our proposed phrase segmentation algorithm can be found in Algorithm 1: SearchPhrase is a function that searches the cached token sequence (i.e., the current candidate for the next phrase) among the most relevant documents. It returns a label that denotes whether the phrase can be found and its position in the relevant documents.

---

[4]Note that the original perplexity of GPT2-Large model on the test set of WikiText-103 is 22.05 (Radford et al., 2019). The gap between it and our results is caused by the different number of samples. In this study, we only use samples that have more than 32 tokens to generate text.

[5]https://huggingface.co/facebook/dpr-ctx_encoder-single-nq-base.

**Algorithm 1:** Phrase Segmentation Algorithm

---

**Data:** Document set: $\mathcal{D} = \{d_i, \{d_j\}_{j=1}^{K}\}_{i=1}^{N}$, where $d_i$ denotes the $i$-th document. $K$ denotes the number of retrieved documents. $N$ denotes the number of documents in the training set. The pre-defined maximum and minimum phrase lengths are $L_{max}$ and $L_{min}$.

**Result:** Segmented document set by phrase granularity: $\mathcal{D}' = \{\{(p_{i,x}, (d_j, \text{pos}_{\text{j}}))\}_{x=1}^{||d_i||_p}\}_{i=1}^{N}$, where $p_{i,x}$ denotes the $x$-th phrase in $d_i$ that also appears in another document $d_j$ in position $j$. $||d_i||_p$ denotes the number of the collected phrases in $d_i$.

**1** **Preprocess**: split each document into token-level pieces by using the off-the-shelf tokenizer.

The preprocessed document set can be formulated as $\mathcal{D} = \{\{t_{i,x}\}_{x=1}^{||d_i||_t}, \{d_j\}_{j=1}^{K}\}_{i=1}^{N}$, where $t_{i,x}$ is the $x$-th token of $d_i$, which consists of $||d_i||_t$ tokens. Prepare the empty list $\mathcal{D}' = \{\}$, empty phrase cache cache$_p$={}, and cached search success label label$_{last}$.

**2** **for** $i \leftarrow 1$ **to** $N$ **do**
**3**      cursor=0
**4**      PhraseCollection={}
**5**      **while** *cursor$\leq ||d_i||_t$* **do**
**6**          **if** $L_{min} \leq$ *len(cache$_p$)* $\leq L_{max}$ **then**
**7**              label$_{now}$, rest=SearchPhrase(cache$_p$)
**8**          **else**
**9**              **if** *len(cache$_p$)* $> L_{max}$ **then**
**10**                  cache$_p$={}
**11**              **end**
**12**          **end**
**13**          **if** *label$_{last}$ is True and label$_{now}$ is False* **then**
**14**              cursor -= 1
**15**              PhraseCollection.append(cache$_p$, rest)
**16**              cache$_p$={}
**17**          **else**
**18**              **if** *label$_{last}$ is False and label$_{now}$ is False* **then**
**19**                  PhraseCollection.append( cache$_p$, None)
**20**                  cache$_p$={}
**21**              **end**
**22**          **end**
**23**          cursor += 1
**24**          label$_{now}$=label$_{last}$
**25**      **end**
**26**      $\mathcal{D}'$.append(PhraseCollection)
**27** **end**

---

# E   MORE CASES

In this section, we present some generated examples of CoG given some specific prefixes. As shown in Figure 4, 5 and 6, it can be observed that the generated continuations are fluent and coherent with the given prefix. However, we also notice some flaws. For example, as shown in Figure 5, CoG copied the phrase *75 mph* from the document ... *sustained winds of at least 120 km / h (75 mph)*, which is incoherent with the previous copied phrase *106 km / h*. Moreover, as shown in Figure 6, CoG copied the phrase *Rhine and Main* from the document (*Bt the terms of the Peace of Basel (22 July 1795), the Prussian army was to leave the Rhine and Main river valleys ...*). However, the complete phrase should be *Rhine and Main river valleys*, and CoG only copy a part of it, leading to inaccurate generation results (*rivers*).

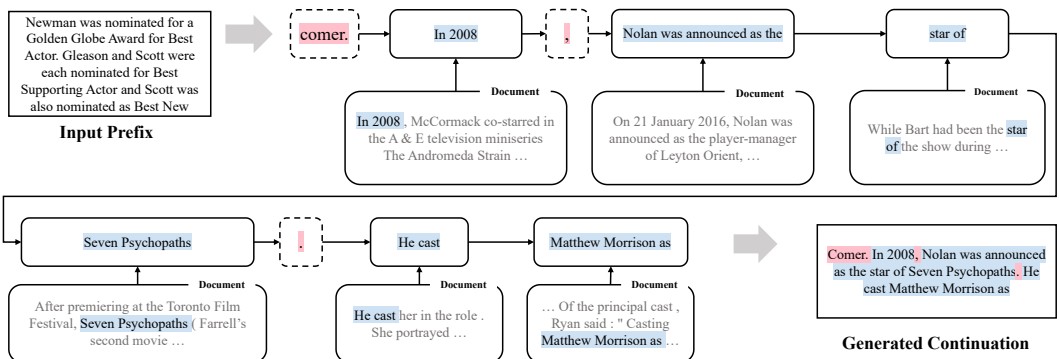

Figure 4: An example generated by CoG. The dotted squares denote that the content (highlighted in red )is generated from the token vocabulary, and the solid squares denote that the content (highlighted in blue ) is copied from other documents.

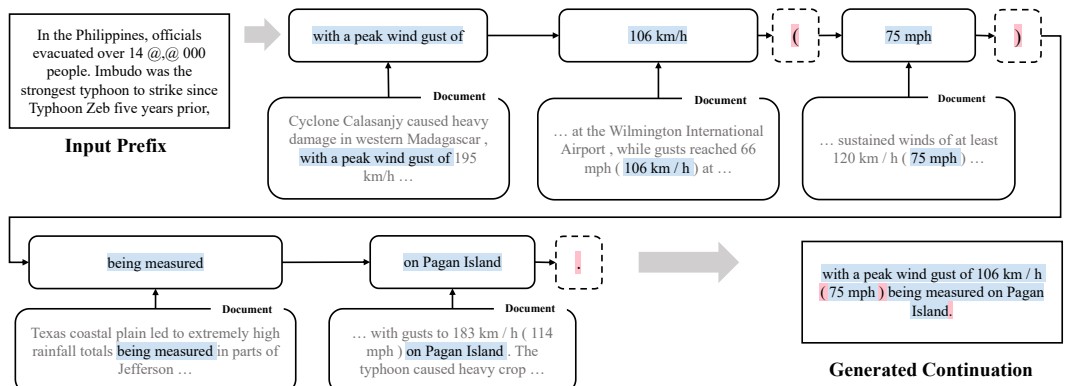

Figure 5: An example generated by CoG. The dotted squares denote that the content (highlighted in red )is generated from the token vocabulary, and the solid squares denote that the content (highlighted in blue ) is copied from other documents.

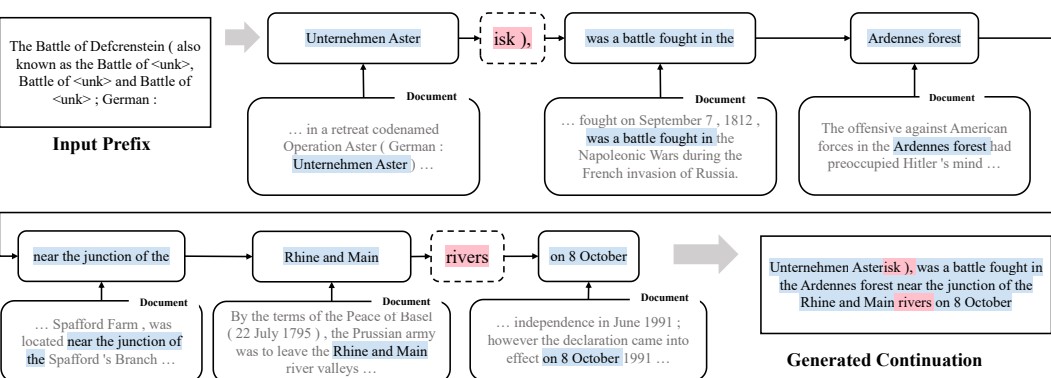

Figure 6: An example generated by CoG. The dotted squares denote that the content (highlighted in red )is generated from the token vocabulary, and the solid squares denote that the content (highlighted in blue ) is copied from other documents.

