# OpenReview forum: "Copy is All You Need"
_ICLR.cc/2023/Conference — ICLR 2023 poster_

### Official Review · Reviewer_pM7y · 2022-10-23

**Confidence:** 4
**Correctness:** 3
**Technical Novelty And Significance:** 3
**Empirical Novelty And Significance:** 3
**Recommendation:** 6

**Clarity, Quality, Novelty And Reproducibility:**

The paper is very well written and the idea is clear. I am not sure of other methods that so simply enables phrase-based vocabulary. I have some questions about method and efficiency. My major concerns are about evaluation with MAUVE which I mentioned as weaknesses.

Is phrase encoder updated every batch?

How large is the "effective" vocab during training vs test time? I suppose this is function of the number of retrieved documents, k = 50.

How do you handle duplicate phrases in the softmax?

I am surprised to hear COPY is faster than vanilla transformer. Sure, can generate more tokens at once, but don't these tokens still need to be processed by the LM after generation? Also, I imagine the size of the softmax is much larger in the COPY approach, although I am not sure. I suppose with K = 50 documents, if there are only a few thousand phrases added to the vocab then maybe it is not such a problem. Along these lines, what size is the word-level vocab used here?

How is it possible to retrieve phrases from local context (like in Fig 2, "Boulter is an")? This was not explained in the main text AFAICT.

One minor comment about the intro... It should be more clear in the intro that Wikitext-103 is not being used in the standard way, and in fact, is not being used as a language modeling benchmark since there is no perplexity eval.

"average inference time cost" --- is it average over words in the test set?

typo: "We denote all the e"

typo: "i is the maximum valid value", I think "i" is not supposed to be the max value here.

nit: Can be confusing that top-k is used in so many different ways. Maybe k with different subscripts will help?

**Strength And Weaknesses:**

# Strengths

- Strong results using MAUVE in all settings (although I find it hard to interpret MAUVE here, and there should be more effort to help readers calibrate their understanding of the reported metrics).

- The method can generate phrases instead of only single words (although I have questions about efficiency).

- The retrieval component uses rich context information.

# Weaknesses

- It is not ideal that COPY uses BERT for retriever and kNN-LM uses vanilla transformer trained from scratch. To me this is a major confounder, and maybe having a strong retrieval component (i.e. BERT) is more important than using phrase-based vocab.

- The rep-N metric is very strict (measures contiguous duplicate n-grams), and perhaps is useful for debugging, but I imagine it does not tell use much about generation quality. It would be more useful to include distribution novel n-grams with respect to ground truth. Can review RAVEN paper for guidance (McCoy et al., 2021. How much do language models copy from their training data? Evaluating linguistic novelty in text generation using RAVEN).

- The results are very reliant on the MAUVE metric, but for wikipedia text, factual correctness is very important and I am not sure how will this is reflected by MAUVE. Although there is much discussion about perplexity being a poor measure of text generation, here, I think it would have been helpful since it could reflect factual correctness to some degree. Unfortunately, authors mentioned that PPL is not reliable because of the phrase-based vocabulary --- I wonder if it would be possible to still compute PPL in some settings if not all (certainly the baselines support PPL at least). In any case, some human eval may be appropriate here so we can better calibrate the interpretation of MAUVE in this setting.

- It would help considerably to have other baselines. This might alleviate concerns about MAUVE too.

**Summary Of The Paper:**

The authors present a retrieval-based LM that can generate phrases. Their base model is a standard transformer, and the retrieval component uses BERT. Their model is trained using in-batch positives / negatives. Retrieval is actually done at document-level, then phrase extracted using a segmentation algorithm. For evaluation, rather than focus solely on language modeling, they evaluate in a text generation setting. They primarily evaluate with MAUVE but also report other metrics useful for analysis. The results are strong compared to baselines, but text generation is very hard to evaluate and I have some questions / suggestions about this.

**Summary Of The Review:**

The work is very interesting and relevant to the community. The main issue is the authors have taken a language modeling task for Wikipedia and converted it into an unconditional text generation task. In unconditional text generation, there are many valid and diverse continuations of text. But in Wikipedia especially I believe that factual information is important to get right, making it hard to interpret some of the evaluation. Human evaluation is one way to immediately address this concern, but perhaps other evaluations besides MAUVE could be useful too. I mentioned perplexity, but maybe BLEU with small sliding window could help understand factual correctness of the model AND leverage the phrase-based aspect.

---

> ### Author Response · Authors · 2022-11-18
> **Response to the Reviewer pM7y**
>
> Thank you for your thoughtful reviews and valuable suggestions!
>
> W1: It is not ideal that COPY uses BERT for retriever and kNN-LM uses vanilla transformer trained from scratch ... a strong retrieval component (i.e. BERT) is more important than using phrase-based vocab.
>
> A: We consider the ability to leverage pre-trained bidirectional language models (e.g., BERT) for creating the phase table as a major advantage of our proposed framework. As described in the first paragraph of Section 4.2, we also attempted to initialize the kNN-LM baseline with pre-trained GPT2 but found that the resulted model can only generate unfluent text. Following previous works [1], we train the GPT2 model from scratch with word-level vocabulary.
>
> W2: The rep-N metric is very strict ... It would be more useful to include distribution novel n-grams with respect to ground truth. Can review RAVEN paper for guidance.
>
> A: Following RAVEN, we have counted the novel n-grams generated by the GPT2, RETRO, and our proposed Copy-Generator models.
>
> | Models | 2-gram novelty | 3-gram novelty | 4-gram novelty | 5-gram novelty | 6-gram novelty |
> | - | - | - | - | - | - |
> | Ground Truth | 0.1120 | 0.3547 | 0.6125 | 0.7717 | 0.853 |
> | Copy-Generator greedy | 0.0780 | 0.2593 | 0.5149 |0.7050 |0.8189|
> | Copy-Generator sampling | 0.0801 | 0.2687 | 0.528 | 0.7199 | 0.8379 |
> | GPT2 greedy | 0.1292 | 0.2622 | 0.4376 | 0.6379 | 0.7915 |
> | GPT2 sampling | 0.1092 | 0.3069 | 0.5687 | 0.7734 | 0.889 |
> | RETRO greedy | 0.066 | 0.1853 | 0.4147 | 0.6431 | 0.7971 |
> |RETRO sampling|0.0835|0.2882|0.5761|0.7849|0.8954|
>
> W3: The results are very reliant on the MAUVE metric, but for wikipedia text ... how will this is reflected by MAUVE. ... I wonder if it would be possible to still compute PPL in some settings if not all. In any case, some human eval ... in this setting.
>
> A: We test the perplexity of generated texts under an external large pre-trained language model (gpt2-large). The results (WikiText-103) are shown below.
> | Models | ppl |
> | - | - |
> | ground truth | 240.33 |
> | gpt2 greedy | 239.97 |
> | gpt2 sampling | 240.16 |
> |RETRO greedy| 240.64 |
> |RETRO sampling | 240.75 |
> |Copy-Generator greedy| 240.54 |
> |Copy-Generator sampling | 240.45|
> It can be observed that perplexity is not a proper metric to evaluate the generation's quality.
>
> W4: It would help considerably to have other baselines. This might alleviate concerns about MAUVE too.
>
> A:We agree that the comparison with more recent baselines can further strengthen our paper. We have added another baseline, RETRO, as suggested by Reviewer siHy. The results of RETRO (on the WikiText-103 test set) are shown below. As seen, RETRO underperforms our approach in terms of MAUVE and diversity scores.
>
> | Models | MAUVE | Diversity | Rep-2 | Rep-3 | Rep-4|
> | - | - | - | - | - | - |
> | RETRO greedy |0.6979 |0.2564 |0.453 |0.3479 |0.281|
> | RETRO sampling|0.6816|0.8171|0.1221|0.0481|0.223|
> |Ours greedy|0.755|0.912|0.049|0.025|0.017|
> |Ours sampling|0.758|0.942|0.039|0.013|0.007|
>
>
> Q1: Is phrase encoder updated every batch?
>
> A: Yes, we fine-tune the last layer of the BERT-based phrase encoder during training.
>
> Q2: How large is the "effective" vocab during training vs test time? I suppose this is a function of the number of retrieved documents, k = 50.
>
> A: Yes, you are right. During training, "effective" vocabulary contains two parts: (1) word-level vocabulary size (50257 in this paper); (2) the phrases in a batch of training documents (256 in this paper). During inference, the "effective" vocabulary consists of the word-level vocabulary and the phrases extracted from the Top-K retrieved documents (K=50 in this paper). The size of the pre-defined word-level vocabulary contains 50257 subwords. Since there are only a few documents encoded to extract the phrase representations, the average number of the phrase representations is 87091.73 in the WikiText-103 test set when K=50.
>
> Q3: How do you handle duplicate phrases in the softmax?
>
> A: Note that duplicate phrases often have different contexts. Therefore, their representations may not be the same and we simply treat them as different phrases.
>
> Q4: I am surprised to hear COPY is faster than the vanilla transformer. ... processed by the LM after generation? Also, I imagine the size of the softmax is much larger in the COPY approach ... what size is the word-level vocab used here?
>
> A: Yes, the tokens in the selected phrases still need to be processed before the next decoding step. However, these tokens could be calculated parallelly, since they have already been generated, which is much faster than the vanilla word-level transformer model. The size of the pre-defined word-level vocabulary contains 50257 subwords. Since there are only a few documents encoded to extract the phrase representations, the average number of the phrase representations is 87091.73 in the WikiText-103 test set when K=50.
>
> [1] Generalization through Memorization: Nearest Neighbor Language Models

---

> > ### Comment · Reviewer_pM7y · 2022-11-18
> > **Quick Question about PPL**
> >
> > Thank you for the detailed response.
> >
> > I noticed that GPT-2 Large has very high perplexity on Wikitext ground truth, but previously it is reported that should be around 22 perplexity. Can you explain the discrepancy? Reference (table 3): https://d4mucfpksywv.cloudfront.net/better-language-models/language-models.pdf

---

> > > ### Author Response · Authors · 2022-11-19
> > > **Response about the high perplexity**
> > >
> > > Thank you very much for pointing out this issue. We indeed did make a mistake in reporting the numbers. We have double-checked the results and updated the numbers. Overall, we believe that PPL is not an ideal metric for quality measurement. As seen, texts generated by greedy search can achieve very low PPL scores (even lower than the ground-truth). This is expected as greedy search targets at likelihood maximization. Sampling-based decoding methods give much higher PPL scores. It is worth noting that Copy-Generator greedy achieves the closest PPL score to ground-truth. The revised results are shown in the following table.
> > >
> > > | Models | ppl |
> > > | - | - |
> > > | ground truth | 25.94 |
> > > | gpt2 greedy | 12.35 |
> > > | gpt2 sampling | 37.04 |
> > > |RETRO greedy| 14.89 |
> > > |RETRO sampling | 48.97 |
> > > |Copy-Generator greedy| 28.47 |
> > > |Copy-Generator sampling | 54.15|

---

### Official Review · Reviewer_LpvV · 2022-10-25

**Confidence:** 3
**Correctness:** 3
**Technical Novelty And Significance:** 3
**Empirical Novelty And Significance:** 3
**Recommendation:** 5

**Clarity, Quality, Novelty And Reproducibility:**

Novel architecture, good and clear writing style.


**Strength And Weaknesses:**

I think that the idea is a reasonable follow-up to the kNN-LM approach of Khandelwal et al. The ability of adding in-domain data without retraining is very appealing. But there are some practical disadvantages compared to vanilla LMs that are brushed over to some extent. First, Copy-Generator does not define a probability distribution over a fixed vocabulary (as the authors admit in Sec. 4.3), and that limits its applicability to pure generation - it does not work in combination with other models (e.g. Bayesian, (re-)scoring...). More importantly, the idea of retrieving phrases from the training data contradicts the trend towards using bigger and bigger training sets. In fact, while latency is reported on Wiki-103 (which has 100M tokens, so rather small by today's standards), it is not reported on the En-Wiki corpus. But even En-Wiki with 3B tokens is not outrageously large nowadays.

A potential way around this latency concern would be a separate treatment of the training set and the document collection used at inference time. A good data point would be the En-Wiki Copy-Generator with an inference-time document collection trimmed down to match the latency of the baselines. More details about the sensitivity regarding the size/domain of the inference-time phrase table would be interesting. I think the note made in Sec. 5.3 "we can use learn COPY-GENERATOR with a smaller corpus but leverage
additional information in a larger corpus in a plug-and-play fashion" misses the point: we want to train with a large corpus, and plug in a small corpus at for inference.

A better sense for the importance of the phrase table would also be provided by reporting Wiki-103 performance of the Copy-Generator with Law-MT phrase table. Is Copy-Generator more/less prone to catastrophic forgetting?

Minor comments:
- It is nice to know that the average length of phrases is 3.95, but more details (variance, full histogram..) would be useful.
- Phrase extraction by segmentation seems limiting as it does not allow any overlap between phrases.
- Why is the Copy-Generator latency for nucleus smaller than for greedy
- Abstract: coping -> copying
- Sec. 3 "We denote all the e all the phrases"
- Sec. 3 "all phrases are context-sensitive" clarify that they depend on the source document context, not on the target prefix context at inference time.
- Sec 4.2: "does not suit work well"

**Summary Of The Paper:**

The proposed Copy-Generator architecture is an attempt to depart from the classical probabilistic token-by-token language modelling perspective for language generation. Instead, generation is framed as gluing together phrases that are retrieved from a collection of documents. Searching for phrases at inference time is implemented in a heuristic coarse-to-fine-grained fashion to reduce the computational cost. The issue of computing a partition function over a potentially ginormous set of phrases is side-stepped by using an NCE loss in training, and either using greedy decoding or nucleus sampling (normalization over top-k items) for inference.

**Summary Of The Review:**

The idea of tackling neural language generation on the phrase-level is neat, but the practical limitations (latency, sensitivity regarding the size and content of phrase table) are not fleshed out enough in the paper.

---

> ### Author Response · Authors · 2022-11-18
> **Response to Reviewer LpvV**
>
> Thank you for your thoughtful reviews and valuable suggestions!
>
> W1: Copy-Generator does not define a probability distribution over a fixed vocabulary
>
> Because the phrase table is not fixed (it depends on the chosen source text collection) and the phrase table can be extremely huge, Copy-Generator does not attempt to compute a well-formed probability distribution. Admittedly, the combination of our approach and other models (e.g. Bayesian, (re-)scoring...) may not be straightforward (or even impossible as suggested by the reviewer). However, we believe the combination is out of the scope of this paper and we leave it as future work.
>
> W2: retrieving phrases from bigger and bigger training sets
>
> First of all, we would like to clarify that, one biggest advantage of our approach is, as described in the Introduction, that our method allows for training-free adaptation to new knowledge sources because the source text collection can be updated in a plug-and-play fashion. In other words, the source text collection during inference does not necessarily need to be the training set. In Section 5.2, the model is trained on a bigger corpus (Wiki-103) but uses a smaller corpus (law-mt) during inference. In Section 5.3, the model is trained on a smaller corpus (Wiki-103) but uses a bigger dataset (En-Wiki) during inference. Additionally, the sensitivity regarding the size of the inference-time phrase table is shown in Figure 3, where we create source text collections of different sizes by using different numbers of documents randomly sampled from En-Wiki. As for the latency concern, we already proposed a coarse-to-fine pipeline as described in the second paragraph in Section 4.2.
>
> W3: Is Copy-Generator more/less prone to catastrophic forgetting
>
> Note that the update of source text collection is done in a plug-and-play fashion and none of the model parameters have been changed. Accordingly, we believe that the situation here is different to the conventional catastrophic forgetting problem (we change the memory wholly and explicitly).
>
> Q: It is nice to know that the average length of phrases is 3.95, but more details (variance, full histogram..) would be useful.
> A: The distribution of the length of copied phrases are shown in the following table. For greedy search decoding methods, the mean and variance of copied phrases' length are 2.389, 2.771 respectively. For sampling decoding method, the mean and variance of copied phrases' length are 2.909, 3.470. It can be found that the sampling method prefers to copy longer phrases than the greedy search method.
>
> |Methods|Uni-gram|2-gram|3-gram|4-gram|5-gram|6-gram|7-gram|8-gram|
> |-|-|-|-|-|-|-|-|-|
> |greedy|0.3834|0.2859|0.1377|0.0704|0.0487|0.0360|0.0236|0.0142|
> |sampling|0.2766|0.2453|0.1734|0.1042|0.0804|0.0598|0.0383|0.0220|
>
>
>
>
>
> Q: Why is the Copy-Generator latency for nucleus smaller than for greedy
>
> A: The main reason for this phenomenon is that the nucleus sampling decoding method samples longer phrases than greedy search, leading to the smaller decoding steps.

---

### Official Review · Reviewer_8gPK · 2022-10-28

**Confidence:** 4
**Correctness:** 4
**Technical Novelty And Significance:** 3
**Empirical Novelty And Significance:** 3
**Recommendation:** 5

**Clarity, Quality, Novelty And Reproducibility:**

The overall quality of this paper is OK. But the authors should further clarify the differences between the proposed method and existing works to highlight their novelty. The reproducibility may be degraded due to the lack of codes.

**Strength And Weaknesses:**

Strengths:

1) Reformulating text generation as a series of actions that fully depend on copying is an interesting idea, which may work well in some NLG scenarios.

2) From the experimental results, copy-generator can achieve better generation quality while having lower latency compared with Transformer-based baselines, which shows its effectiveness.

3) This paper is well-organized and easy to follow.

Weaknesses:

1) The idea which combines vanilla token-level generation and phrase copying has been already discussed in the existing works such as [1]. In my view, the main difference only falls into the format of knowledge sources (unstructured documents in this paper and structured knowledge in [1]). Thus, the authors should further clarify the difference between copy-generator and existing works to show their novelty.

2) The metrics used in the experiment are somewhat questionable. First, this paper lacks the fluency metric such as perplexity. From my experience, if copy operations dominate the generation process compared with token-level generation, the fluency of the whole sentence may be degraded. Then, MAUVE as a distribution-based metric may be biased to evaluate copy-generator because copy-generator largely copies source texts. If the distributions of source texts and the test set are similar enough, it’s not surprising that copy-generator can achieve a high MAUVE score.

3) The latest baseline selected in this paper is in 2020. The authors should add state-of-the-art baselines about retrieval-augmented language models such as [2].

4) Since the experiment is mainly conducted on WikiText-103, I wonder whether there is a principle to collect the source texts for a specific downstream task / dataset, such as machine translation, text summarization or dialogue generation. It’s important because the generation quality of copy-generator largely depend on the quality of source texts.

5) Typo: “all the e” in the second paragraph of Section 3 should be removed.

[1] Latent Relation Language Models. AAAI 2020.

[2] GNN-LM: Language Modeling based on Global Contexts via GNN. ICLR 2022.


**Summary Of The Paper:**

This paper formulates text generation as copying from existing text segments. The authors decompose text generation as a series of copy-and-paste operations by seeking text spans from existing articles and copying them as the generation result at each step. Experimental results on WikiText-103 show that the proposed method can achieve higher MAUVE scores and also support effective domain adaption and better inference efficiency.

**Summary Of The Review:**

Although generating texts via a series of copy operations is interesting, I’d recommend the authors to solve the concerns about the novelty and experimental settings before making this paper ready for publication.

---

> ### Author Response · Authors · 2022-11-18
> **Response to reviewer 8gPK**
>
> Thank you for your thoughtful reviews and valuable suggestions!
>
> W1: The idea which combines vanilla token-level generation and phrase copying has been already discussed in the existing works such as [1]. In my view, the main difference only falls into the format of knowledge sources (unstructured documents in this paper and structured knowledge in [1]). Thus, the authors should further clarify the difference between copy-generator and existing works to show their novelty.
>
> A: Thanks for pointing out the reference. We would like to clarify three major differences between our work and [1]. First, our method allows the copy of any phrases while [1] only allows copying entity names in a knowledge graph. Second, [1] focused on entity-centred explanation generation (in their work, an entity must be pre-specified about which the model generates a description). In contrast, we target at open-ended text generation, where the model can generate continuation for any given prefix. Last but not least, as the reviewer pointed out, [1] uses knowledge graphs as knowledge source while ours uses plain text corpus. We believe this actually makes a huge difference. Knowledge graph is known to suffer from sparsity, requires large human labour, and is easy to become outdated if not maintained timely. In contrast, textual corpora contain rich and diverse information, and is easy to collect/scale (think of the success of pre-trained language models).
>
>
> W2: The metrics used in the experiment are somewhat questionable. First, this paper lacks the fluency metric such as perplexity. From my experience, if copy operations dominate the generation process compared with token-level generation, the fluency of the whole sentence may be degraded. Then, MAUVE as a distribution-based metric may be biassed to evaluate copy-generator because copy-generator largely copies source texts. If the distributions of source texts and the test set are similar enough, it’s not surprising that a copy-generator can achieve a high MAUVE score.
>
> A: Thanks for your constructive suggestion. We test the perplexity of generated texts under a large pre-trained language model (gpt2-large). The results are shown below.
> | Models | ppl |
> | - | - |
> | ground truth | 25.94 |
> | gpt2 greedy | 12.35 |
> | gpt2 sampling | 37.04 |
> |RETRO greedy| 14.89 |
> |RETRO sampling | 48.97 |
> |Copy-Generator greedy| 28.47 |
> |Copy-Generator sampling | 54.15|
> Overall, we believe that PPL is not an ideal metric for quality measurement. As seen, texts generated by greedy search can achieve very low PPL scores (even lower than the ground-truth). This is expected as greedy search targets at likelihood maximization. Sampling-based decoding methods give much higher PPL scores. It is worth noting that Copy-Generator greedy achieves the closest PPL score to ground-truth.
>
> W3: The latest baseline selected in this paper is in 2020. The authors should add state-of-the-art baselines about retrieval-augmented language models such as [2].
>
> A: Thank you very much for pointing out the baseline issue. We very much agree that the comparison with more recent baselines can further strengthen our paper. Because [2] is built upon kNN-LM and only gives marginal improvements, we instead choose another baseline, as suggested by Reviewer siHy, RETRO. The results of RETRO (on the WikiText-103 test set) are shown below. As seen, RETRO underperforms our approach in terms of MAUVE and diversity scores.
>
> |Models|MAUVE|Diversity|Rep-2|Rep-3|Rep-4|
> |-|-|-|-|-|-|
> |RETRO greedy|0.6979|0.2564|0.453|0.3479|0.281|
> |RETRO sampling|0.6816|0.8171|0.1221|0.0481|0.223|
> |Ours greedy|0.755|0.912|0.049|0.025|0.017|
> |Ours sampling|0.758|0.942|0.039|0.013|0.007|
>
> W4: Since the experiment is mainly conducted on WikiText-103, I wonder whether there is a principle to collect the source texts for a specific downstream task / dataset, such as machine translation, text summarization or dialogue generation. It’s important because the generation quality of copy-generator largely depend on the quality of source texts.
> A: We believe the general principle of selecting source text collection is to choose (1) texts with similar domains/topics of the intended output (2) text of high quality in general. Particularly, for machine translation, the best choice would be high-quality in-domain corpus in the target language. For summarization/dialogue, we would suggest also including the input document/dialogue to the source collection.

---

### Official Review · Reviewer_siHy · 2022-10-29

**Confidence:** 4
**Correctness:** 3
**Technical Novelty And Significance:** 4
**Empirical Novelty And Significance:** 3
**Recommendation:** 8

**Clarity, Quality, Novelty And Reproducibility:**

#### Clarity
If I understand this paper properly, the pipelined approach presented in Section 4.2 as 'implementation details' are essential during both inference and training. In particular, I believe that the BERT phrase encoder parameters are only ever updated for the small set of retrieved documents, and there is no need to re-index the phrase table during training (the phrase representation is operating more like a re-ranker). If my understanding is correct, then I suggest this detail is moved out from a paragraph focusing on 'inference efficiency' into the description of the core approach. Without this detail, the description of the training procedure is very hard to understand.

Otherwise, this paper is well written and easy to follow.

#### Reproducability
More should be said about the construction of the InfoNCE loss. How exactly are the in-batch negatives constructed? Are these all phrases in all retrieved documents for all prefixes in the batch? Also, what happens when there are multiple separate 'correct' representations of the target phrase $p_k$ coming from different contexts in the retrieved documents?


#### Novelty & Quality
Phrase encoding and retrieval has been studied before, as has contextualized token representation retrieval for language modeling. But, I believe this is the first work to do phrase retrieval for language modeling, or any type of long-form generation. That seems significant.


**Strength And Weaknesses:**

#### Strengths
- This is a novel and interesting architecture that is significantly different to previous work.
- Copy generator performs well in comparison to reasonable baselines.
- While the model is only applied to language modeling tasks, I can imagine extensions to other forms of generation where retrieving and copying text is likely to be a good strategy (e.g. multi-document summarization).

#### Weaknesses
- The paper relies heavily on the recently proposed MAUVE metric. It would be great to also have some human judgement of quality, no matter how small, to get more intuitions of where Copy-generator does better or worse than the baselines.
- Copy-generator makes use of context-dependent phrase encodings. However, the examples in Figures 2 and 5-8 seem to show that large phrases are being drawn from very different contexts to the target context. It'd be good to see some analysis of how important the is here.
- It would be interesting to see a comparison to retrieval augmented language models (e.g. RETRO, ATLAS) which also include document retrieval at inference time (but also have a greater inference-time computational cost than copy-generator).


**Summary Of The Paper:**

This paper presents a language model that can autoregressively choose tokens from a fixed vocabulary, or multi-token phrases from a pre-encoded corpus. The paper builds on previous work on phrase encoding for question answering, but extends this work to freeform text generation. The model (Copy-generator) is evaluated on three separate language modeling tasks. Two that use Wikipedia text and one that involves domain transfer to a legal domain. When compared to a vanilla LM and a KNN-LM that retrieves memories of individual tokens, Copy-generator performs better according to the recently proposed MAUVE metric. It does less well, in comparison to KNN-LM, according to diversity and Rep-N metrics on some tasks, though. Traditional perplexity scores are not particularly meaningful here because the model can choose different segmentations of the text than those used by the baselines.

The phrase encoder is trained along with the language model in the following fashion. First a greedy segmentation algorithm is run to choose a single segmentation of each training example into phrases that exist elsewhere in the training corpus. Then, for each phrase, a piplined retrieval system is run. First a document retrieval system retrieves a small set of documents that are similar to the prefix. Then, a phrase search is applied over all phrases in this retrieved set of documents. Phrases are represented using encodings of their start and end position, following previous work on phrase retrieval. The phrase retrieval model and language model are trained using a combination of InfoNCE loss over phrase candiates, as well as language model loss. The document retrieval model is fixed (as far as I can make out).

At inference time, phrases or individual tokens are selected from the union of all phrases in a large retrieval corpus and the fixed length vocabulary. Examples show that Copy-generator often chooses to select long phrases instead of individual tokens (I would like to also see statistics illustrating how often different length n-grams occur), and the generated text is deemed significantly better than the baseline systems according to the automatic MAUVE metric, which has been trained to correlate with human judgements. An analysis of performance shows that quality increases with the number of documents in the phrase corpus, and this increase is particularly pronounced when greedy search is used during decoding.


**Summary Of The Review:**

This paper presents a novel approach to language modeling, that can copy large phrases from a retrieval corpus rather than decoding them one token at a time. The performance on three language modeling datasets are positive, according to the MAUVE metric and in relation to the similar KNN-LM baseline that performs retrieval of single token encodings. It would be nice to see some more analysis of the outputs, and how the copy mechanism is being used during inference time. It also would be very interesting to see how well this model does on a wider variety of text generation tasks. However, the current paper is sufficiently novel and interesting for acceptance at ICLR.

---

> ### Author Response · Authors · 2022-11-19
> **Response to Reviewer siHy**
>
> Thank you very much for your insightful and encouraging review!
>
> W1:The paper relies heavily on the recently proposed MAUVE metric. It would be great to also have some human judgement of quality, no matter how small, to get more intuitions of where Copy-Generator does better or worse than the baselines.
>
> A: We conducted human evaluation to compare our Copy-Generator (Copy-Generator greedy) against the conventional autoregressive language model (GPT2 sampling). We ask human annotators to choose which one is better. On 50 randomly selected prefixes, the numbers of win/tie/loss cases are 21, 12, and 17, respectively.
>
> W2: Copy-generator makes use of context-dependent phrase encodings. However, the examples in Figures 2 and 5-8 seem to show that large phrases are being drawn from very different contexts to the target context. It'd be good to see some analysis of how important this is here.
>
> A: For document retrieval, we employ a semantic retriever (SimCSE). On the one hand, we find that the retrieved documents are relevant in terms of topic/domain (e.g., television, show, film in Figure 2). On the other hand, we find that entity mismatch is a prevalent problem, likely due to the fact that our retriever does not rely on lexical overlapping. We leave the investigation of more "accurate" document retrieval as future work.
>
> W3: It would be interesting to see a comparison to retrieval augmented language models (e.g. RETRO, ATLAS) which also include document retrieval at inference time (but also have a greater inference-time computational cost than copy-generator).
>
> A: Thanks for pointing out the additional baselines. We agree that the comparison with the recent baselines can further strengthen our paper. We report the results of RETRO on the WikiText-103 test set as below. As seen, RETRO underperforms our approach in terms of MAUVE and diversity scores.
>
> | Models         | MAUVE  | Diversity | Rep-2  | Rep-3  | Rep-4 |
> | -------------- | ------ | --------- | ------ | ------ | ----- |
> | RETRO greedy   | 0.6979 | 0.2564    | 0.453  | 0.3479 | 0.281 |
> | RETRO sampling | 0.6816 | 0.8171    | 0.1221 | 0.0481 | 0.223 |
> | Ours greedy    | 0.755  | 0.912     | 0.049  | 0.025  | 0.017 |
> | Ours sampling  | 0.758  | 0.942     | 0.039  | 0.013  | 0.007 |
>
> Below, we also provide answers to the additional questions mentioned in your review.
>
> Q1: I would like to also see statistics illustrating how often different length n-grams occur
>
> A: Thanks for your constructive suggestion. We counted the numbers of copied phrases of different lengths (on the WikiText-103 test set), and the statistics are shown in the following table. It can be observed that longer n-grams have lower frequencies.
> | Methods  | Uni-gram | 2-gram | 3-gram | 4-gram | 5-gram | 6-gram | 7-gram | 8-gram |
> | -------- | -------- | ------ | ------ | ------ | ------ | ------ | ------ | ------ |
> | greedy   | 0.3834   | 0.2859 | 0.1377 | 0.0704 | 0.0487 | 0.0360 | 0.0236 | 0.0142 |
> | sampling | 0.2766   | 0.2453 | 0.1734 | 0.1042 | 0.0804 | 0.0598 | 0.0383 | 0.0220 |
>
> Q2: In particular, I believe that the BERT phrase encoder parameters are only ever updated for the small set of retrieved documents, and there is no need to re-index the phrase table during training (the phrase representation is operating more like a re-ranker).
>
> A: At each training iteration, the BERT phrase encoder encodes the retrieved documents for generating phrase representations. In other words, we do not maintain a phrase table containing all the phrases in the training corpus but compute a small subset on-the-fly.
>
> Q3: More should be said about the construction of the InfoNCE loss. How exactly are the in-batch negatives constructed? Are these all phrases in all retrieved documents for all prefixes in the batch? Also, what happens when there are multiple separate 'correct' representations of the target phrase coming from different contexts in the retrieved documents?
>
> A: Referring to the second paragraph in Section 3.2. The in-batch negatives come from all other phrases in the retrieved documents. The "correct" phrase for a given prefix is pre-defined by the phrase segmentation algorithm described in Section 3.2 and Appendix A. It is indeed possible that the target phrase may appear multiple times in the retrieved document but currently we only take one as the positive.

---

### Decision · Program_Chairs · 2023-01-20

**Decision:**

Accept: poster

**Justification For Why Not Higher Score:**

I have concerns about the evaluation of the method, and believe that more careful evaluation should be conducted to make the paper stronger.

**Justification For Why Not Lower Score:**

The paper propose a novel and interesting research direction, and some reviewers expressed a strong interest in building on top of the method.

**Metareview: Summary, Strengths And Weaknesses:**

In this paper, the authors propose a new method for text generation, relying on retrieval and copy of chunks of tokens that are relevant to continue the text. The method is inspired by retrieval augmented LM, such as KNN-LM, but introduces the following technical contributions: (1) the method retrieves full chunks, meaning that it can generates multiple tokens in one step (2) a new training algorithm, based on contrastive learning.
The reviewers enjoyed this paper, as it opens a new research direction for text generation. They found the paper well written, and believe that this method will lead to interesting follow-up works. However, there is a large concern over evaluation of the method, which mostly rely on the recently proposed automatic MAUVE metric. I am really on the fence for this paper, as I believe that ICLR should publish exploratory work, even if the evaluations are a bit weak. However in this case, I also believe that the authors could have done more to convince readers that the method is promising, by reporting more automatic metrics, and performing more careful human evaluations. I also believe that additional baselines, which copy large amount of text, should be included in the automatic metric.

**Note From Pc:**

if the above contains the word "oral" or "spotlight" please see: "oral" presentation means -> notable-top-5% and "spotlight" means -> notable-top-25%. As stated in our emails, we are disassociating presentation type from AC recommendations